# Capsinoids Increase Antioxidative Enzyme Activity and Prevent Obesity-Induced Cardiac Injury without Positively Modulating Body Fat Accumulation and Cardiac Oxidative Biomarkers

**DOI:** 10.3390/nu16183183

**Published:** 2024-09-20

**Authors:** Késsia Cristina Carvalho Santos, Lucas Furtado Domingos, Fabiane Merigueti Nunes, Luisa Martins Simmer, Evellyn Rodrigues Cordeiro, Filipe Martinuzo Filetti, Danilo Sales Bocalini, Camila Renata Corrêa, Ana Paula Lima-Leopoldo, André Soares Leopoldo

**Affiliations:** 1Postgraduate Program in Nutrition and Health, Health Sciences Center, Federal University of Espírito Santo, Vitória 29075-910, ES, Brazil; kecarvalhosantos@gmail.com (K.C.C.S.); domingos.lucass@gmail.com (L.F.D.); nunesmerigueti@gmail.com (F.M.N.); luisamsnutricionista@gmail.com (L.M.S.); ana.leopoldo@ufes.br (A.P.L.-L.); 2Postgraduate Program in Physiological Sciences, Health Sciences Center, Federal University of Espírito Santo, Vitória 29075-910, ES, Brazil; evellyn_rc@hotmail.com (E.R.C.); filipe.m.filetti@hotmail.com (F.M.F.); 3Postgraduate Program in Physical Education, Physical Education and Sports Center, Federal University of Espírito Santo, Vitória 29075-910, ES, Brazil; bocaliniht@hotmail.com; 4Medical School, São Paulo State University (UNESP), Botucatu 18618-686, SP, Brazil; camila.camacho@unesp.br

**Keywords:** capsinoids, cardiac remodeling, oxidative stress, antioxidant capacity, obesity

## Abstract

Background/Objectives: Capsinoids are potential antioxidant agents capable of reducing oxidative damage and the resulting complications triggered by obesity. Thus, this study aimed to investigate the effects of capsinoids on adiposity and biomarkers of cardiac oxidative stress in obese rats induced by a high-fat diet. Methods: Male *Wistar* rats were exposed to a high-fat diet for 27 consecutive weeks. After the characterization of obesity (week 19), some of the obese animals began to receive capsinoids (10 mg/kg/day) by orogastric gavage. Adiposity and comorbidities were assessed. In the heart, remodeling, injury, and biomarkers of oxidative stress were determined. Results: The treatment did not reduce obesity-induced adiposity but was efficient in reducing cholesterol levels. Capsinoid treatment did not cause a difference in heart and LV mass, despite having reduced troponin I concentrations. Furthermore, capsinoids did not reduce the increase in the advanced oxidation of protein products and carbonylated proteins caused by obesity in cardiac tissue. In addition, obese rats treated with capsinoids presented high levels of malondialdehyde and greater antioxidant enzyme activity compared to untreated obese rats. Conclusions: In conclusion, treatment with capsinoids increases antioxidative enzyme activity and prevents obesity-induced cardiac injury without positively modulating body fat accumulation and cardiac oxidative biomarkers.

## 1. Introduction

Obesity, defined as the excessive accumulation of body fat, is considered a risk factor for impairment of health and mortality in the population [1,2,3,4,5,6]. This disease affects around 810 million people and is related to more than 10 million deaths around the world, making it an important global public health problem [5].

In obesity, the imbalance between energy consumption and expenditure promotes hypertrophy and hyperplasia of white adipose tissue (WAT) [7,8,9]. However, the stimulus of the expansion of these adipocytes drives the entry of cells from the immune system, influencing the construction of an inflammatory and dysfunctional profile of WAT [9]. These changes may increase morbidity biomarkers, such as increased reactive species, the induction of insulin resistance and hyperleptinemia, decreased glucose tolerance, lipotoxicity, ectopic lipid deposition, and inflammation [7,8,9]. In the context of cardiovascular diseases (CVDs), these effects are relevant, as they are related to a greater risk of developing left ventricular hypertrophy, hypertension, cardiometabolic disorders, atrial fibrillation, dysfunction, and heart failure [9,10,11,12,13].

Furthermore, as previously stated, obesity is related to an increase in oxygen reactive species (ROS) in the organism, such as superoxide radicals (O_2_^•−^), hydrogen peroxide (H_2_O_2_), hydroxyl radicals (•OH), nitric oxide radicals (NO•), and peroxynitrite (ONOO^−^) [14]. The body is capable of naturally producing such species [15], which at controlled levels are related to the control of bodily homeostasis and contribute to processes such as immunological defense, cellular proliferation and differentiation, defense against pathogens, and signaling, among others [14,16,17]. However, some external sources, such as high-calorie diets, can increase and exacerbate ROS, triggering toxic and harmful effects on the body and the oxidative stress state [14,15].

Oxidative stress is a state of imbalance between oxidants and antioxidants in favor of oxidants, which leads to a dysregulation of signaling and redox control and/or molecular damage [18]. Studies highlight that excessive fat accumulation contributes to the release of pro-oxidants and proinflammatory cytokines in the organism [1,7]. In this way, the overstimulation of ROS production and release can cause changes in protein structures and functions, damage to nucleic acids and lipids, carcinogenesis, and inflammation [14,15].

Specifically, regarding changes occurring in the heart, obesity causes a reduction in metabolic flexibility and a decrease in cardiac efficiency [19]. In this context, the increase in circulating free fatty acids can increase the mitochondrial uncoupling and reduce the efficiency of oxygen use from this substrate, causing a deficiency in energy production and the increased release of reactive species in cardiac tissue [19]. In turn, reactive species can lead to cardiac hypertrophy and dysfunction, becoming relevant for the development of myocardial dysfunction and consequent heart failure, among other cardiovascular complications [19,20].

In this context, as well as the high incidence of morbidity and mortality and the high cost of obesity treatment and CVDs, searching for alternatives for the treatment and prevention of these diseases becomes relevant [21,22]. Thus, the literature has searched for alternatives for the treatment and prevention of obesity and its complications. In this scenario, studies with capsinoids (CAPs) have been used; CAPs consist of alkaloid molecules that are vanillic alcohol esters with fatty acids that are present in peppers of the *Capsicum* genus [23,24,25]. The CAP group includes compounds such as capsiate, dihydrocapsiate, and nordihydrocapsiate [23,25]. CAPs have demonstrated effects such as promoting thermogenesis, stimulating energy and lipid metabolism, promoting body mass loss, and reducing adipose tissue. Furthermore, they can improve glucose tolerance, have antihyperlipidemic activity, and have anti-inflammatory and antioxidant properties [23,25,26,27]. Regarding antioxidant activity, the literature has shown that compounds belonging to the capsinoids group can exhibit activities such as the restoration of the gene expression of antioxidant enzymes, protection against lipid oxidation, and reduction of 8-hydroxyguanosine (8-OHdG), a marker of oxidative damage in proteins and DNA [28,29]. They may confer a protective and mediating effect against oxidative stress.

Thus, there is evidence that capsinoids may have effects on obesity and its consequences. However, most studies do not take into account obesity associated with CVD. Therefore, considering the physiological properties of CAPs and the absence of studies on cardiac stress’s effect on obesity, the current study was performed to test the hypothesis that the chronic administration of capsinoids positively modulates biomarkers of oxidative stress in the heart in obesity, resulting in a reduction in products derived from oxidative damage and the prevention of pathological cardiac remodeling.

## 2. Materials and Methods

### 2.1. Animal Care and Experimental Protocol

After approval by the ethics committee for the use of animals at the Federal University of Espírito Santo, Brazil (08/2022), male Wistar rats (n = 44) were subjected to a protocol of 27 consecutive weeks, which was divided into two moments: exposure to experimental diets for 19 weeks (obesity induction and maintenance) and treatment with capsinoids for 8 weeks. Initially, the rats were randomized into two groups: SD (standard diet) and HFD (high-fat diet). The standard diet was based on the AIN 93 recommendations, being composed of corn starch, casein, dextrinized starch, sucrose, soy oil, microcrystalline cellulose, mineral mix, vitamin mix, L-cystine, BHT, and bitartrate choline (Pragsoluções biociências^®^, São Paulo, Brazil). This diet was composed of 9.45% lipids (100% soy oil), 75.66% carbohydrates, and 14.89% proteins (Pragsoluções biociências^®^, São Paulo, Brazil). For the composition of the high-fat diet, the same recommendations were followed, but with the addition of lard (Pragsoluções biociências^®^, São Paulo, Brazil). The high-fat diet was composed of 45.33% lipids (11.5% soy oil and 88.5% pork lard), 40.29% carbohydrates, and 14.38% proteins (Pragsoluções biociências^®^, São Paulo, Brazil).

During the 27-week protocol, 40 g of food and water were offered daily ad libitum. Each diet’s energy density was also important for estimating daily caloric intake [30] and feed efficiency (FE) (FE = total body weight gain of the animals (g)/total energy ingested [kcal]) [31,32].

The characterization of obesity occurred when the HFD group showed a significant increase in body weight (BW) in relation to the SD group. This moment was demonstrated in previous studies [33,34]. After the initial moment of obesity (week 16), the SD and HFD groups were renamed the control (C) and obese (Ob) groups, respectively (Figure 1). Afterward, the rats were maintained in a state of obesity for 4 weeks. At the end of the 19th week of the protocol, the rats were randomized into four different groups according to the absence and/or presence of capsinoid treatment: C (standard diet), CCap (standard diet with capsinoids), Ob (obese), and ObCap (obese with capsinoids) (Figure 1). However, considering that this study aimed only to evaluate the effects of capsinoid treatment on obesity, the CCap rats were excluded from the current study and used in other studies performed in our laboratory (Figure 1).

### 2.2. Administration of Capsinoids

The ObCap group was supplemented daily with capsinoids (CAPs; Infinity Pharma, Brazil) (capsiate, dihydrocapsiate, and nordihydrocapsiate) by orogastric gavage (10 mg of capsinoids/kg of BW diluted in 1 mL of water/kg of BW) for 8 weeks. The capsinoid dose was adjusted weekly according to the change in BW to maintain a constant CAP dose throughout the study. The C and Ob groups received vehicle gavage in the same amounts.

### 2.3. Euthanasia

At the end of the experimental protocol (27 weeks), following a 6-hour fasting period, the animals were heparinized and anesthetized with a solution containing ketamine hydrochloride (90 mg/kg) and xylazine hydrochloride (10 mg/kg) [35]. In cases where animals still exhibited signs of nociceptive reflex after anesthetic induction, an anesthetic overdose (lethal dose) was administered, consisting of three times the doses of ketamine hydrochloride and xylazine hydrochloride used during the animals’ anesthetic induction [35]. Following euthanasia, the animals were submitted to a median thoracotomy to collect blood and tissue samples.

### 2.4. Body Weight and Adiposity

To characterize the rats’ obesity, their body weight, body fat, and adiposity index (AI) were analyzed. In addition, the evolution of body mass was measured weekly. The amount of body fat was determined through the sum of epididymal, retroperitoneal, and visceral fat pads. Finally, the AI was calculated by the formula AI = amount of body fat/final BW multiplied by 100 [36,37].

### 2.5. Visceral Adipose Tissue Morphology

For visceral adipose tissue morphometry, collected samples were fixed in 4% paraformaldehyde. They were then dehydrated in ethanol, clarified in xylene and embedded in paraffin. After inclusion, sections with a thickness of 5 μm were obtained using a rotating microtome (Leica^®^ RM 2125 RTS, Wetzar, Germany) and placed on slides for histology. Then, the histological sections were stained with hematoxylin and eosin (H&E). For adipocyte area analysis, 10 fields from each adipose tissue slide were checked. The morphometric analysis was performed blindly. And the images were captured using a 10× objective lens, with the aid of a video camera (LAS EZ^®^, ICC50 HD—51112061, Leica, Germany) coupled to an optical microscope (Leica^®^, RM 2125 RTS, Wetzar, Germany). After capturing the images, Image J Pro-Plus^®^ 4.5 (Media Cybernetics, Silver Spring, MD, USA) was used to identify and convert the area of adipocytes considering the scale in μm in each field of the slides.

### 2.6. Comorbidities Associated with Obesity

To evaluate the possible metabolic and hormonal changes induced by obesity, a glucose tolerance test (GTT), a homeostatic model assessment for the insulin resistance index (HOMA-IR), and lipid and hormonal profiles were evaluated.

In the last week of the protocol (week 27), the animals were subjected to a glucose tolerance test. They remained fasting for 6 h to analyze glycemic levels in basal conditions and after glucose overload (50% glucose; i.p.) at 30, 60, 90, and 120 min [38]. The area under the curve (AUC) for glucose was also evaluated to identify the presence or absence of glucose tolerance.

Homeostatic model assessment for the HOMA-IR was used to determine insulin resistance, calculated by the following formula: fasting insulin concentration (μU/mL) × fasting blood glucose (mmol/L)/22.5 [39].

For lipid and hormonal profiles analysis, blood samples were collected in Falcon tubes and centrifuged at 10,000 rpm for 10 min and then stored at −80 °C. The insulin (Elabscience Biotechnology Inc., Houston, TX, USA), leptin (R&D Systems, Minneapolis, MN, USA), adiponectin (Elabscience Biotechnology Inc., Houston, TX, USA) and glucagon (R&D Systems, Minneapolis, MN, USA) ware determined by enzyme-linked immunosorbent assay (ELISA) using specific kits. Plasmatic concentrations of triglycerides (TG), total cholesterol (TC), high-density lipoproteins (HDL) and low-density lipoproteins (LDL) was determined using specific kits (Bioclin^®^, Belo Horizonte, Brazil) and analyzed with the BS-200 automated biochemical apparatus (Mindray, Shenzhen, China).

### 2.7. Cardiac Remodeling and Injury

The cardiac remodeling process was evaluated by analyzing the total mass of the heart and the left ventricle (LV) and their relationship with tibia length [40]. Cardiac injury was measured by troponin I concentration, determined using a specific ELISA kit (AFG Scientific LLC, Northbrook, IL, USA).

### 2.8. Analysis of the Production of Reactive Species

The determination of ROS was carried out using the dihydroethidium (DHE) fluorescence method, which makes it possible to analyze the “in situ” production of the superoxide radical. Transverse sections of the left ventricle (8 μm) were placed on gelatinized slides. Afterward, the slides were washed and incubated with Krebs’s solution in a light-protected (2 μM DHE) humidified chamber at 37 °C for 30 min to detect superoxide. The intensity of fluorescence was detected at 585 nm using a confocal fluorescence microscope (Leica DM 2500 TI, Nikon Instruments Inc., Melville, NY, USA). After capturing the fluorescence images, specific software (Image J Pro-Plus^®^ 4.5, Media Cybernetics, Silver Spring, MD, USA) was used to quantify the fluorescence intensity on the slides and, consequently, to quantify the production of reactive species in the left ventricle. Quantification was performed in arbitrary units, and 4 fields were analyzed per sample.

### 2.9. Biomarkers of Cardiac Oxidants and Antioxidants

Fragments of LV were homogenized in 1:10 sodium phosphate buffer (SPB) and transferred to Eppendorf microcentrifuge tubes. Then, the samples were centrifuged at 3500 rpm at 4 °C for 10 min.

The supernatant was used to measure oxidant biomarkers, such as malondialdehyde (MDA), carbonyl proteins (CBO), advanced oxidation protein products (AOPP), and antioxidant biomarkers, such as the antioxidant capacity (FRAP) and the activity of the enzymes catalase (CAT) and superoxide dismutase (SOD).

To evaluate the peroxidation of membrane lipids, the concentration of MDA was assessed using the thiobarbituric acid reactive substances method (TBARS). For this purpose, the homogenates were centrifuged together with a solution containing thiobarbituric acid, trichloroacetic acid 2M, hydrochloric acid, and distilled water for 10 min at 25 °C. The supernatants were collected and boiled for 45 min. Then, the reading was carried out at 532 nm and 600 nm. The MDA concentration was obtained, taking into account the difference between the absorbances. The concentration of MDA was obtained through the molar extinction coefficient (1.56 × 10^5^ M^−1^ cm^−1^) and the absorbances of the samples, and the final result was expressed in nmol/g of protein.

CBO was measured using the 2,4-dinitrophenylhydrazine (DNPH) derivatizer and spectrophotometric detection of proteins modified by the carbonylation process [41]. To this end, solutions containing DNPH and HCL (2M) and another with NaOH and distilled water were prepared. For the analyses, the homogenate diluted in a solution containing DNPH and HCL (2M) was used. Afterward, the samples were incubated for 10 min at room temperature; then, NaOH solution (6 M) and distilled water were added. Another 10 min of incubation at room temperature while being protected from light was necessary until reading at 450 nm in a microplate reader, with the result obtained through the following calculation: [(sample abs./22,000) × 1,000,000]/abs. of the protein. AOPP was measured using a technique that uses potassium iodide (1.16 M), glacial acetic acid, and a curve prepared with chloramine, with a reading at 340 nm.

The antioxidant capacity was measured using the ferric reducing antioxidant power (FRAP) technique, using a working reagent that contained acetate buffer, TPTZ/HCl solution, and ferric chloride. In a plate, the working solution was added to the sample. The reading was performed at 594 nm after 30 min of waiting.

The CAT activity was evaluated by following the decrease in the levels of hydrogen peroxide. For analysis, the samples were diluted in 10% Triton solution. This preparation was left on ice for 15 min to release the catalase. Then, 10 mM potassium phosphate buffer was added at room temperature. Then, the medium with hydrogen peroxide was added. After 2 min, a reading was carried out at 240 nm for 3’30”, reading every 30 s.

To determine the activity of the SOD, glycine and catalase buffer were added to the sample wells. Afterward, a solution with adrenaline and HCL PA was added. Immediately afterward, a reading was taken at 412 nm every 40 s for 20 min.

### 2.10. Statistical Analysis

The data distribution was assessed using the Kolmogorov–Smirnov normality test. The results are expressed as mean ± standard deviation. All comparisons (SD vs. HFD, as well as C vs. Ob and Ob vs. ObCap) were performed using Student’s *t*-test for independent samples. However, to analyze the evolution of body mass during the period of exposure to experimental diets, a two-way ANOVA for independent samples was used, complemented by Bonferroni’s post hoc test. During the capsinoid treatment period, a two-way ANOVA for independent samples was used for body mass and GTT evaluation, complemented by Tukey’s post hoc test. The significance level adopted was 5%. The statistical analyses and graphics were conducted and created using GraphPad Prism 9.0 software (GraphPad, San Diego, CA, USA).

## 3. Results

### 3.1. Exposure to Experimental Diets

Figure 2 demonstrates the evolution of body mass during the period of exposure to the experimental diets. As indicated, from the 16th week onward, the HFD group (601 ± 71 g) presented a significantly higher (+8.5%) BW than the SD group (554 ± 56 g), characterizing the initial moment of obesity. The period between the 16th and 19th weeks was called the period of obesity maintenance, and the groups were renamed C and Ob, respectively. At the end of this period (19th week), it was verified that the final body weight (FBW) was higher in the Ob group (606 ± 67 g) compared to C (551 ± 53 g), representing an elevation of 10%. However, there was no significant difference in body mass gain between the groups (C: −2.68 ± 14.13 g; Ob: 4.39 ± 17.60 g).

### 3.2. Exposure to Capsinoid Treatment

Considering the nutritional profile, during the chronic administration of capsinoids, the Ob group (15.1 ± 1.5 g/day) presented a reduction (−22.2%) in food consumption when compared to the C group (19.4 ± 2.4 g/day), but this parameter was similar to that of the ObCap group (15.3 ± 2.2 g/day). In addition, no significant differences were observed between Ob and ObCap in relation to caloric intake (Ob: 72.8 ± 7.3 kcal/day vs. ObCap: 73.6 ± 10.7 kcal/day) and feed efficiency (Ob: −0.29 ± 0.96% vs. ObCap: 0.08 ± 0.91%), respectively.

Regarding the body weight at week 19, the results indicate that the IBW was higher in Ob rats than in C (+9%; C: 544 ± 50 g vs. Ob: 593 ± 51 g), but the FBW was similar (C: 543 ± 43 g vs. 583 ± 51 g; *p* > 0.05). Additionally, there were no statistical differences between the Ob and ObCap groups for IBW (Figure 3A) and for FBW after 8 weeks of capsinoid treatment (Figure 3B), respectively, indicating that the animals’ body mass gain stabilized (Ob = ObCap, *p* > 0.05).

Finally, the results showed that Ob rats presented an increase in body adiposity relative to C, as shown by high values for all parameters, such as epididymal (C: 8.3 ± 1.9 g vs. 11.6 ± 1.8 g), retroperitoneal (C: 12.6 ± 1.6 g vs. Ob: 20.6 ± 4.8 g), visceral (C: 6.8 ± 1.7 g vs. 11.7 ± 2.3 g), and body fat (C: 28.4 ± 3.4 g vs. Ob: 43.9 ± 7.9 g), demonstrating the high-fat diet’s efficiency in developing obesity (Ob > C, *p* < 0.05). Nevertheless, the capsinoid treatment could not reduce or prevent the accumulation of total fat and increase in fat pads in obesity since ObCap animals presented higher values than Ob for epididymal fat (+15.5%), retroperitoneal fat (+34.5%), and total body fat (+23.5%) (Figure 3C,D,F). However, when total body fat values were normalized by body weight, the results showed that there was no difference between Ob and ObCap (Figure 3G).

Furthermore, in relation to visceral adipose tissue, no difference was observed in adiposity area among groups, revealing the existence of heterogeneous adipocytes with regard to their size in both groups (Figure 4). In addition, the Ob visceral adipocyte area was similar to that of C (C: 120 ± 25 µm^2^ vs. Ob: 141 ± 19 µm^2^).

Table 1 illustrates the lipid, glycemic, and hormonal profiles after treatment with capsinoids. The results showed that the Ob group had 69.3% higher cholesterol levels relative to the C group (C: 41.8 ± 9.0 mg/dL vs. Ob: 70.8 ± 6.8 mg/dL). In addition, there were no differences between C and Ob for the other metabolic parameters (TG, HDL, and LD). One important result is related to the effect of capsinoid treatment, which was able to reduce cholesterol in obesity (−18.8%) (ObCap < Ob, *p* < 0.05) without promoting positive changes in TG, HDL, and LDL levels (Table 1).

Regarding glycemic parameters, the results showed that the baseline glucose obtained from GTT was increased (15.7%) in ObCap relative to Ob rats, but the glycemic values after the administration of glucose overload were similar among groups. However, there were no alterations in the AUC for glucose between the groups (Table 1). The insulin levels were elevated in Ob compared to C rats (+128%; C: 35.4 ± 9.0 pg/mL vs. Ob: 80.8 ± 19.9 pg/mL; *p* > 0.05); however, treatment with capsinoids did not prevent this increase (*p* > 0.05) (Table 1). The same behavior was observed in relation to the HOMA-IR, in which the Ob group presented values 144% higher than the C group (C: 0.23 ± 0.07 vs. Ob: 0.57 ± 0.14), indicating insulin resistance induced by obesity. However, treatment with capsinoids could also not improve this behavior.

Regarding the hormonal profile, an increase in leptin levels was found in the Ob group (+60.5%) when compared to the C group (C: 1.81 ± 0.60 ng/mL vs. Ob: 2.90 ± 0.30), but no statistical differences were observed in glucagon and adiponectin. In relation to capsinoid treatment, there were no statistical differences in leptin, glucagon, and adiponectin levels between the Ob and ObCap groups (Table 1).

Table 2 summarizes the cardiac injury and remodeling after treatment with capsinoids. The results indicate that the Ob group presented an increase in heart (+17.8%; C: 1.28 ± 0.13 g vs. Ob: 1.51 ± 0.17) and LV (+37.2%; C: 0.72 ± 0.12 g vs. Ob:0.99 ± 0.22) in relation to the C group, respectively. However, after normalization by tibia length, this difference did not remain, with no difference between the groups. However, it is worth noting that capsinoids treatment was not able to prevent the cardiac remodeling induced by obesity.

Despite that, the results showed that treatment with capsinoids for obesity was able to reduce the troponin I (−33%) in the ObCap group when compared to the Ob group, indicating that the treatment triggered a reduction in this marker of cardiac injury (Table 2).

Furthermore, the production of superoxide radicals in situ in LV was verified using DHE as a marker. Obesity did not promote alterations in superoxide formation relative to the C group (C: 10.4 ± 2.8 a.u. vs. Ob: 10.9 ± 2.6 a.u.; C = Ob, *p* > 0.05). In addition, there was also no difference between the obese groups for this parameter (Figure 5).

Moreover, when the cardiac biomarkers of oxidative stress and antioxidant enzymes were checked, the data showed no difference between Ob and C groups for MDA, a marker of lipid peroxidation (C: 21.8 ± 5.9 nmol/mg protein vs. Ob: 21.1 ± 9.5 nmol/mg protein; *p* > 0.05). However, MDA levels in ObCap rats were higher than in the Ob group (+76.9%), indicating that the treatment negatively affected lipid peroxidation in the animals’ hearts (Figure 6A). In addition to this finding, the levels of CBO (C: 1.09 ± 0.37 nmol/mg protein vs. Ob: 3.97 ± 1.38 nmol/mg protein) and AOPP (C: 2.90 ± 0.83 μmol/L per Chloramine T unit/mg protein vs. Ob: 6.00 ± 0.94 μmol/L per Chloramine T unit/mg protein) were increased in the Ob group in relation to the C group (+265% and +76.8%, respectively); however, no difference was observed compared to ObCap (Figure 6B,C). The results also show a statistical difference in antioxidant capacity (FRAP) between C and Ob (C: 5.41 ± 2.04 nmol Fe/mg vs. Ob: 9.32 ± 1.87 nmol Fe/mg) but no difference between the Ob groups (Figure 6D). These findings highlight that treatment with capsinoids was not efficient in reversing oxidative changes and damage triggered by obesity.

Finally, it was demonstrated that obesity was unable to promote significant changes in the antioxidant capacity in relation to the C group. However, SOD and CAT were elevated in ObCap compared to Ob (+62.5% and +636.4%, respectively) (Figure 6E and Figure 6F, respectively). Thus, these results showed that capsinoids are potentiating a climate of elevated ROS.

## 4. Discussion

This study’s main findings were that chronic treatment with capsinoids in obesity could not reduce body weight, prevent excessive accumulation of adiposity, or significantly reverse metabolic changes. Specifically, in the heart, it was observed that treatment with capsinoids was able to reduce the cardiac injury marker and elevate the activity of antioxidant enzymes. However, the chronic administration of capsinoids was unable to reverse the oxidative stress induced by obesity. Furthermore, treatment with capsinoids promoted greater lipid peroxidation in the animals’ hearts.

Experimental models of obesity have been extensively used in research to study the response to various forms of the disease [42]. In experiments where obesity is induced by a high-fat diet, progression to the obese phenotype tends to be a slow process, taking 16–20 weeks for a significant increase in the animals’ body mass [42]. Additionally, the literature indicates that diets that provide 30–60% of total calories from fat favor the development of obesity and associated comorbidities [43,44]. In the current study, obesity was characterized from the 16th week onward, with an increase in body weight in the HFD group when compared to the SD group. Previous studies from our laboratory have already shown a bodyweight increase when using a high-fat diet for 20 weeks [33,34]. However, research with a shorter intervention time (8 weeks) has also managed to observe the efficiency of different high-fat diets in increasing body mass [45,46].

Considering body fat pads, our results demonstrate that the high-fat diet induced an increase in epididymal, visceral, retroperitoneal, and total body fat compared to the group that received a standard diet, characterizing obesity. However, despite CAP emerging as a compound with potential action for controlling and reducing obesity [3,4,47,48,49], in the current study, the treatment with capsinoids was not efficient in reducing body fat; on the contrary, they increased these values. However, there was no alteration in the adiposity index. These results indicate the inefficiency of the treatment in reducing the obesity phenotype, which suggests that this effect may be related to the increase in circulating leptin in the treated group [50]. Furthermore, despite the increase in adiposity, when the morphology of visceral adipose tissue was verified, there was no difference in the adipocyte area between the groups, suggesting that the increase in adipose mass was driven by adipocyte hyperplasia instead of hypertrophy.

Moreover, taking into account the biochemical and physiological disorders related to the disease, in a previous study carried out with enzymatically synthesized capsiate and dihydrocapsiate, the treatment’s effect was to reduce serum cholesterol and serum lipids in animals [51]. In line with this, in this research, although obesity only induced isolated hypercholesterolemia in animals, a beneficial effect of capsinoids was observed to reverse this condition. A possible explanation for this effect would be that capsinoids may have similar effects to their capsaicinoid analogs, which can reduce HMG-CoA reductase mRNA and increase CYP7A1 mRNA in the liver, decreasing cholesterol synthesis and increasing their conversion to bile acids, respectively [52]. However, specific studies on the effects of capsinoids on lipid metabolism are necessary to confirm this assumption.

Additionally, in obesity, the release of leptin becomes highly stimulated, leading to a state of leptin resistance, altering the regulation of appetite and energy expenditure [8,9,53]. Furthermore, adiponectin secretion tends to be reduced, triggering effects such as increased inflammation and blood triglyceride levels [8,54,55]. In disagreement, in the current study, leptin resistance induced by obesity was not observed. In addition, the diet and/or exposure time were not efficient in causing changes in serum adiponectin levels. Furthermore, no benefit was observed in reducing leptin and improving these hormonal parameters when animals were subjected to treatment with capsinoids. One of the reasons capsinoids did not change these hormones’ patterns of secretion is that the treatment did not reduce body fat mass, so the increase in tissue could continue to stimulate the distorted secretion of leptin and adiponectin [50]. Another important aspect is that the increased insulin levels in the group treated with capsinoids may have increased the expression and release of leptin [50].

Furthermore, it is worth noting that obesity can trigger direct or indirect changes in cardiac function [10,13]. In this context, increased body adiposity may induce cardiovascular changes like an increase in volume and pressure overload, which often triggers concentric hypertrophy of the LV, dilation, and degradation of the tissue matrix, and cardiac fibrosis, leading to heart dysfunction and failure [10,13]. In addition, the ectopic accumulation of lipids also induces the release of pro-atherogenic factors, can cause stress on the heart wall, and can lead to myocardial injury, triggering concentric hypertrophy, LV remodeling, and cardiac failure [12,13]. Linked to the changes mentioned, in heart failure, it is also possible to observe an increase of cardiac troponin, a marker of myocardial injury that can increase as a consequence of myocardial stretching resulting from volume overload, subendocardial ischemia, elevated intracardiac pressure, and/or arrhythmias [56].

Considering this context, it was expected that the hearts of obese rats would present hypertrophy. However, although the hearts and LV of obese animals showed an increase in mass, when the values were normalized by the tibia length, no difference was observed. Given these consequences, treatment with capsinoids also did not produce any effect in reducing the mass of the heart or ventricle. These effects may be due to the inefficiency of the diet or the time to induce comorbidities associated with obesity, even though increases in adiposity, plasma cholesterol, and insulin resistance were observed in Ob animals. Gasparini et al. [57] also did not observe changes in the mass of the heart and ventricle in obese rats after 20 weeks of obesity induction, even though they used a high-fat diet. However, Cavalera, Wang, and Frangogiannis [10] indicate that genetic models of obesity may be more effective in inducing fibrosis and cardiac hypertrophy. The authors also mention that in models of obesity induced by high-fat diets, the establishment of cardiac fibrosis may be late [10]. Furthermore, the severity of these changes may depend on variables of the obesity model, such as the species, age of the animals, mechanism of obesity, and presence of other pathophysiological conditions [10].

Additionally, considering the effects of capsinoids on cardiac remodeling, Wang et al. [49] previously demonstrated that transient receptor potential vanilloid subfamily member 1 (TRPV1) can mediate the hypertrophic response triggered by pressure overload in the heart and that the anti-hypertrophic action of its analog capsaicin depends precisely on the presence of TRPV1 channels. This effect appears to be mediated by the induction of PPAR-δ expression and the inhibition of NF-κB [58]. In this regard, despite the first being more potent, it is known that both capsaicin and the capsinoid group are TRPV1 receptor agonists [23,25,27]. However, unlike capsaicin, capsinoids are quickly hydrolyzed in the gastrointestinal tract, possibly hindering the subsequent activation of TRPV1 receptors present in other tissues, such as the heart [24,28]. Thus, it is worth noting that the effects of TRPV1 activation on cardiac hypertrophy have not yet been completely established, and more studies are needed to understand this issue [49].

Moreover, the model of obesity was not efficient in inducing myocardial injury. However, the treatment with capsinoids decreased the concentration of this marker. Horiuchi et al. [59] previously identified that in humans, a higher body mass index (BMI) was associated with lower troponin levels compared to overweight or normal-weight people; however, the reasons for this effect are still poorly understood. It is worth mentioning that in our study, although BMI was not checked in the animals, the obese group treated with capsinoids showed increased values of body adiposity, which could be a possible explanation for the low troponin I levels in the treated group. Related to the reduction of troponin I by CAP, it has already been demonstrated that TRPV1 knockout mice had higher plasma troponin concentrations after myocardial infarction compared to animals in which TRPV1 was not silenced [60]. In this way, it is suggested that the reducing effect of capsinoids on this marker of cardiac injury is possibly related to the expression and activation of TRPV1 in the heart.

In addition, several studies have demonstrated that obesity increases the oxidative stress on the heart [19,20,61]. For instance, in obesity and associated comorbidities, greater production of hydrogen peroxide and superoxide in cardiac cells has been observed [12]. Also, a reduction in the expression and/or activity of antioxidant enzymes in the heart and circulation of people with obesity is commonly observed [19]. These effects cause consequences in the cardiac structure, size, and function, constituting a damaging process for the myocardium [12,20,61]. Regarding the oxidative damage induced by obesity and the potential treatment with capsinoids and their analogs, it was demonstrated that capsiate and its analogs can promote protection against lipid oxidation [29]. It was also found that the activation of TRPV1 by its agonists can increase the expression of uncoupling protein 2 (UCP2), important for reducing the generation of reactive species in the electron transport chain in mitochondria [62]. In addition, TRPV1 can exert a protective effect on the myocardium by being able to reduce markers of endogenous peroxynitrite formation [58].

In relation to oxidative biomarkers, in our study, no increase in superoxide levels by DHE was observed in obese animals, and no changes were observed due to treatment with capsinoids. This result may be due to the rapid conversion of superoxide to hydrogen peroxide inside the cells, which has a longer life cycle and permeates cell environments better, making it capable of causing persistent cellular changes, such as lipid and membrane oxidation and protein oxidation [61,63,64].

In this context, our results indicate the inefficiency of the capsinoid treatment in reducing carbonyl proteins and advanced oxidation protein products caused by obesity. Some reasons for these effects may be related to these findings, including the dosage, exposure to treatment, and vehicle. Furthermore, it may be that the antioxidant portion of the capsinoid molecule (suggested to be the phenol portion) is not targeted or does not have easy access to ROS binding sites in the myocardium [20]. An important finding, different from what was expected, was that treatment with capsinoids caused greater lipid peroxidation in the hearts of obese animals, observed by the increase in MDA. This result suggests that compensatory mechanisms related to chronic antioxidant exposure in ROS-producing systems may be involved [20]. We also suggest that somehow, the treatment, or even the activation of the TRPV1 receptor in the heart, induced an increase in enzymatic lipid peroxidation, in which enzymes such as lipoxygenases (LOX), cyclooxygenases (COX), and cytochrome P450 (CYP450) synthesize molecules capable of triggering peroxidation [64].

Furthermore, our organism has antioxidant systems that work to maintain the balance of these molecules and reactions [20,61]. For example, reactive species are neutralized by intracellular antioxidant enzymes such as superoxide dismutase (SOD) and catalase (CAT) [61]. SOD constitutes a group of metalloenzymes that dismutate superoxide into hydrogen peroxide [64,65], which is important for maintaining an adequate flow of hydrogen peroxide [65]. Meanwhile, CAT constitutes a group of enzymes responsible for the decomposition of peroxide [20,64]. Our results also showed that the antioxidant capacity carried out by FRAP was elevated in obesity without observed differences with CAP treatment. This finding indicates that oxidative alterations induced by obesity somehow provoked the antioxidant response to counterbalance the damage. Furthermore, the increase in biomarkers in obesity was not accompanied by an elevation in antioxidant enzymes’ activity, which was already expected. However, it is worth highlighting that treatment with capsinoids promoted an increase in antioxidant activity visualized by the increases in SOD and CAT activities. One possible explanation is that this result may not necessarily be an isolated protective effect of capsinoids on the heart but may be a consequence of increased lipid peroxidation. In this sense, the literature reports that lipid peroxidation products are capable of inducing the transcription of nuclear factor erythroid 2-related factor 2 (Nrf2), important for regulating the expression of antioxidant enzymes [19], which could promote protective feedback against oxidative damage itself.

## 5. Conclusions

Treatment with capsinoids does not prevent body fat accumulation or promote improvements in the obesity-induced metabolic profile, nor does it suppress the size of visceral fat. Specifically in heart tissue, capsinoid supplementation does not prevent the process of cardiac remodeling and does not lead to the positive modulation of oxidative stress in high-fat-diet-induced obesity. However, treatment with capsinoids causes a reduction in troponin I and induces antioxidant activity in cardiac tissue.

## Figures and Tables

**Figure 1 nutrients-16-03183-f001:**
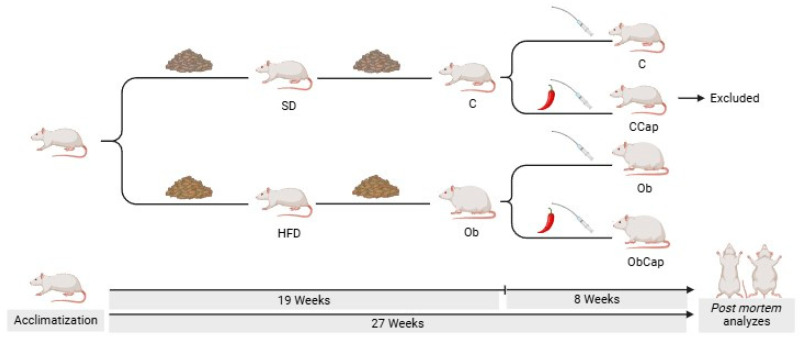
Schematic representation of experimental protocol (27 weeks). SD: standard diet (n = 17); HFD: high-fat diet (n = 27); C: control (n = 11); CCap: standard diet with capsinoids (n = 6); Ob: obese (n = 13); ObCap: obese with capsinoids (n = 14).

**Figure 2 nutrients-16-03183-f002:**
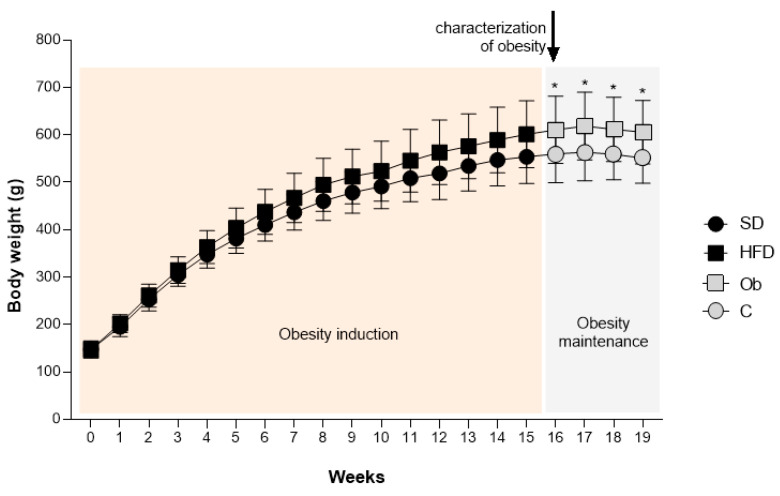
Evolution of body mass during the period of exposure to experimental diets. Groups SD: standard diet (n = 17); HFD: high-fat diet (n = 27). After the onset and characterization of obesity, the groups were renamed: Groups C: control (n = 17), Ob: obese (n = 27). Values are expressed as mean ± standard deviation. Two-way ANOVA for independent samples complemented with Bonferroni post hoc test—* *p* < 0.05 Ob vs. C.

**Figure 3 nutrients-16-03183-f003:**
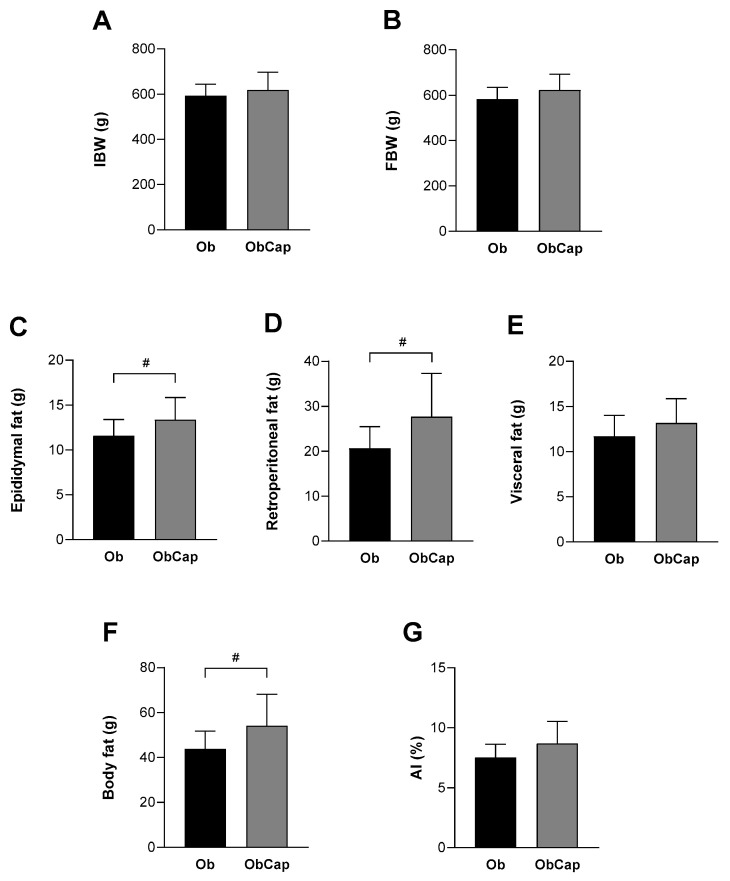
Body weight and adiposity profile after treatment with capsinoids. (**A**) IBW: initial body weight. (**B**) FBW: final body weight. (**C**–**E**): fat pads. (**F**): body fat. (**G**): AI: adiposity index. Groups Ob: obese (n = 13); ObCap: obese treated with capsinoids (n = 14). Values are expressed as mean ± standard deviation. Student *t*-test for independent samples—^#^ *p* < 0.05—ObCap vs. Ob.

**Figure 4 nutrients-16-03183-f004:**
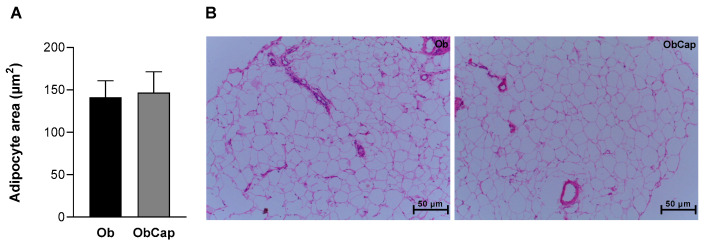
Adipocytes of visceral adipose tissue. (**A**) Adipocyte area of visceral fat pad. (**B**) Representative histological images of visceral fat pad of groups stained with hematoxylin and eosin. The scale bar is 50 μm. Groups Ob: obese (n = 6); ObCap: obese treated with capsinoids (n = 7). Values are expressed as mean ± standard deviation. Student *t*-test for independent samples.

**Figure 5 nutrients-16-03183-f005:**
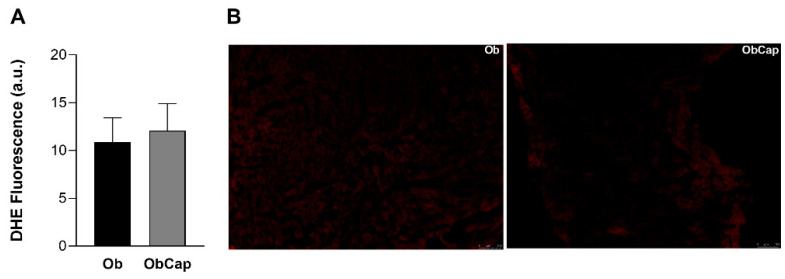
Analysis of superoxide formation in sections of cardiac tissue by the dihydroetidium fluorescence (DHE). (**A**) DHE Fluorescence. (**B**) Representative images of LV with DHE fluorescence. The scale bar is 100 μm. Groups Ob: obese (n = 6); ObCap: obese treated with capsinoids (n = 7). Values are expressed as mean ± standard deviation. Student *t*-test for independent samples.

**Figure 6 nutrients-16-03183-f006:**
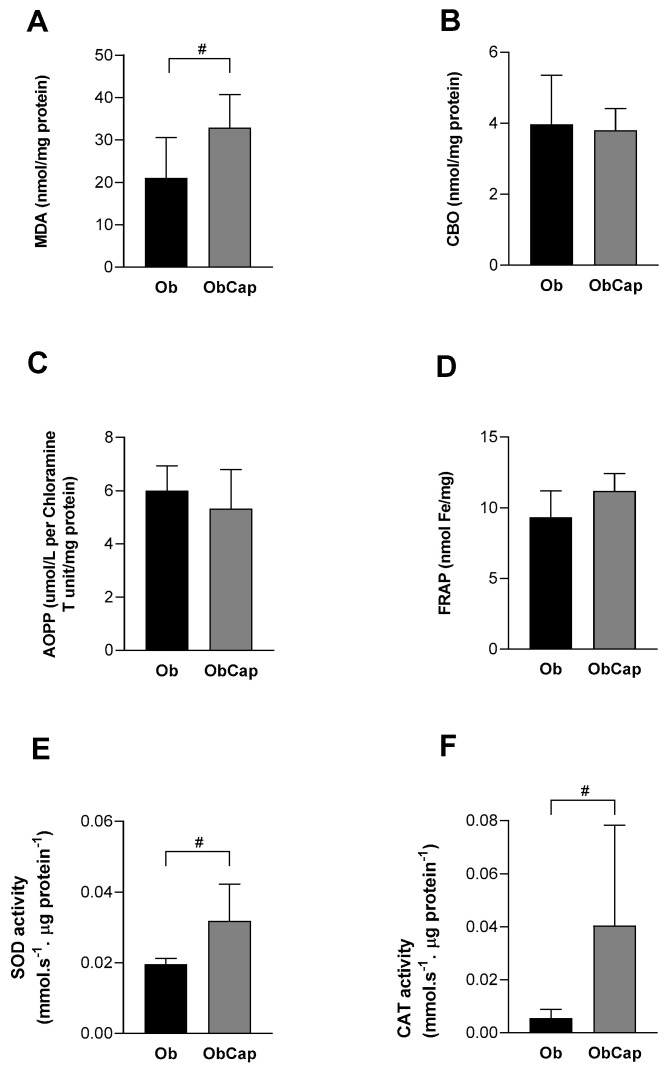
Effect of capsinoid treatment on cardiac biomarkers of oxidative stress and on antioxidant enzymes. (**A**) MDA: malondialdehyde. (**B**) CBO: carbonyl proteins. (**C**) AOPP: advanced oxidation protein products. (**D**) FRAP: iron-reducing antioxidant capacity. (**E**) SOD: superoxide dismutase. (**F**) CAT: catalase. Groups Ob: obese (n = 6); ObCap: obese treated with capsinoids (n = 7). Values are expressed as mean ± standard deviation. Student *t*-test for independent samples -^#^ *p* < 0.05-ObCap vs. Ob.

**Table 1 nutrients-16-03183-t001:** Lipid, glycemic, and hormonal profiles after treatment with capsinoids.

Variable	Groups	
	Ob	ObCap
**Cholesterol (mg/dL)**	70.8 ± 6.8	57.5 ± 18.2 ^#^
**Triglycerides (mg/dL)**	32.1 ± 17.1	19.5 ± 6.4
**HDL (mg/dL)**	18.2 ± 3.3	17.5 ± 5.3
**LDL (mg/dL)**	13.1 ± 2.7	12.9 ± 3.3
**Glucose (mg/dL)**	102 ± 8	118 ± 7 ^#^
**AUC for glucose (mg/dL · min)**	943 ± 106	1071 ± 178
**Insulin (pg/mL)**	80.8 ± 19.9	73.1 ± 11.3
**HOMA-IR**	0.57 ± 0.14	0.62 ± 0.09
**Leptin (ng/mL)**	2.90 ± 0.30	3.03 ± 0.38
**Glucagon (ng/mL)**	0.12 ± 0.04	0.11 ± 0.01
**Adiponectin (ng/mL)**	43.5 ± 10.1	45.5 ± 5.5

HDL: high-density lipoprotein. LDL: low-density lipoprotein. AUC: area under the curve for glucose. HOMA-IR: homeostatic model assessment for insulin resistance. Groups Ob: obese; ObCap: obese treated with capsinoids. Leptin (Ob: n = 7, ObCap: n = 7); Glucagon and Adiponectin (Ob: n = 6, ObCap: n = 7); Cholesterol, triglycerides, HDL e LDL (Ob: n = 9, ObCap: n = 9); Blood glucose (Ob: n = 8, ObCap: n = 10); Insulin e HOMA-IR (Ob: n = 7, ObCap: n = 5). Values are expressed as mean ± standard deviation. Student *t*-test for independent samples—^#^ *p* < 0.05—ObCap vs. Ob.

**Table 2 nutrients-16-03183-t002:** Cardiac injury and remodeling process after treatment with capsinoids.

Variable	Groups	
	Ob	ObCap
**Heart (g)**	1.51 ± 0.17	1.47 ± 1.10
**Hearth/tibia length (g/cm)**	0.37 ± 0.03	0.33 ± 0.02
**LV (g)**	0.99 ± 0.22	1.06 ± 0.07
**LV/tibia length (g/cm)**	0.22 ± 0.05	0.24 ± 0.01
**Troponin I (pg/mL)**	18.46 ± 3.28	12.37 ± 2.79 ^#^

LV: left ventricle. Groups Ob: obese (n = 6); ObCap: obese treated with capsinoids (n = 7). Values are expressed as mean ± standard deviation. Student *t*-test for independent samples—^#^ *p* < 0.05—ObCap vs. Ob.

## Data Availability

The datasets used and/or analyzed during the current study are available from the corresponding author upon reasonable request.

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
