# Peer review of "Capsinoids Increase Antioxidative Enzyme Activity and Prevent Obesity-Induced Cardiac Injury without Positively Modulating Body Fat Accumulation and Cardiac Oxidative Biomarkers"

_nutrients, 2024, doi:10.3390/nu16183183_

Round 1

Reviewer 1 Report

Comments and Suggestions for Authors

Manuscript Number: 3142743

Journal: Nutrient

Title: Capsinoids promotes an increase in enzyme antioxidative activity and prevents the injury cardiac induced by obesity without positively modulate the cardiac oxidative biomarkers

Comments:

In this paper, researcher investigate the effects of capsinoid treatment on biomarkers of oxidative stress in the hearts of rats with obesity induced by high-fat diet. The main findings were that chronic treatment with capsinoids in obesity was not able to reduce body weight and to prevent excessive accumulation of adiposity, as well as significantly to reversing the metabolic changes. Specifically in the heart, it was observed that treatment with capsinoids was able to prevent the cardiac injury and to elevate the antioxidant enzymes in the heart. In addition, the chronic administration of capsinoids was unable to reverse the oxidative stress induced by obesity. Furthermore, thetreatment with capsinoids promoted greater lipid peroxidation in the hearts of animals belonging to the ObCap group. However, significant revisions should be made before publication is considered, and the following are the main observations:

1. How were the doses of capsaicin determined? Why were multiple concentration gradients not set up but only one set chosen?

2. The fluorescence signal is not visible in the fluorescence picture in Figure 5.

3. Why did the rats in the ObCap group have higher MDA levels than the Ob group, while SOD and CAT were elevated in the ObCap group compared to the Ob group? How can this result be explained?

4. The conclusion section is too succinct; it is suggested to be added.

5. References are not introduced in a uniform format, e.g. line 597, line 232.

6. The discussion section repeats the introduction of too much literature, suggest streamlining.

7. Capsaicinoids are potential antioxidants that can reduce obesity-induced oxidative damage and its complications. Why do the results suggest that capsaicin treatment presents the opposite effect in reducing obesity-induced oxidative stress?

8. The secondary headings in the Materials and Methods section are not sequenced.

9. There are many writing irregularities in the article, such as H2O2 should be revised to H2O2 and O2- to O2- in line 59

Author Response

Manuscript number:

Nutrients-3142743

Title:

Capsinoids promotes an increase in enzyme antioxidative activity and prevents the injury cardiac induced by obesity without positively modulate the cardiac oxidative biomarkers

REVIEWER COMMENTS

Reviewer 1

Journal: Nutrient

Title: Capsinoids promotes an increase in enzyme antioxidative activity and prevents the injury cardiac induced by obesity without positively modulate the cardiac oxidative biomarkers

Comments:

In this paper, researcher investigate the effects of capsinoid treatment on biomarkers of oxidative stress in the hearts of rats with obesity induced by high-fat diet. The main findings were that chronic treatment with capsinoids in obesity was not able to reduce body weight and to prevent excessive accumulation of adiposity, as well as significantly to reversing the metabolic changes. Specifically in the heart, it was observed that treatment with capsinoids was able to prevent the cardiac injury and to elevate the antioxidant enzymes in the heart. In addition, the chronic administration of capsinoids was unable to reverse the oxidative stress induced by obesity. Furthermore, thetreatment with capsinoids promoted greater lipid peroxidation in the hearts of animals belonging to the ObCap group. However, significant revisions should be made before publication is considered, and the following are the main observations:

  1. How were the doses of capsaicin determined? Why were multiple concentration gradients not set up but only one set chosen?

Dear Reviewer, we appreciate the commentary, and as requested we explain below why we chose the dose of capsinoids in the current study. The choice of dose was based on previous studies showing that an anti-obesity effect was observed at a minimum dose of 10 mg of capsinoids/kg of BW diluted in 1 ml of water/kg of body weight (1-5). It should be noted that we used capsinoids that contained capsiate in their formulation, a non-pungent capsaicin analog. Several studies have related the benefits of capsinoids or capsiate on obesity, including the reduction of body fat and anti-obesity effects (1-5). In addition, the most of studies have been using the capsaicin or capsiate in diet (3,5), administered orally (1,2) and via stomach tube (4); in our study the daily supplementation of capsinoids (CAP; Infinity Pharma, Brazil) was performed with orogastric gavage. This situation may have influenced our findings, as well as the dose used and the exposure of treatment. Moriyama et al. (2003) reported that the capsiate has little pungency and its oral stimulation did not induce nociceptive responses, unlike capsaisin, i.e., the oral cavity may not be a target for capsinoids to enhance the energy expenditure and the intragastric administration of capsiate via a stomach tube would be increased, but this fact was not observed in the current study (6).

  1. Haramizu S, Kawabata F, Ohnuki K, Inoue N, Watanabe T, Yazawa S, Fushiki T. Capsiate, a non-pungent capsaicin analog, reduces body fat without weight rebound like swimming exercise in mice. Biomed Res. 2011;32(4):279-84. doi: 10.2220/biomedres.32.279.
  2. Haramizu S, Kawabata F, Masuda Y, Ohnuki K, Watanabe T, Yazawa S, Fushiki T. Capsinoids, non-pungent capsaicin analogs, reduce body fat accumulation without weight rebound unlike dietary restriction in mice. Biosci Biotechnol Biochem. 2011;75(1):95-9. doi: 10.1271/bbb.100577. Epub 2011 Jan 7.
  3. Hong Q, Xia C, Xiangying H, Quan Y. Capsinoids suppress fat accumulation via lipid metabolism. Mol Med Rep. 2015;11(3):1669-74. doi: 10.3892/mmr.2014.2996. Epub 2014 Nov 24.
  4. Ohnuki K, Haramizu S, Oki K, Watanabe T, Yazawa S, Fushiki T. Administration of capsiate, a non-pungent capsaicin analog, promotes energy metabolism and suppresses body fat accumulation in mice. Biosci Biotechnol Biochem 2001;65(12):2735-40. doi: 10.1271/bbb.65.2735.
  5. Okamatsu-Ogura Y, Tsubota A, Ohyama K, Nogusa Y, Saito M, Kimura K. Capsinoids suppress diet-induced obesity through uncoupling protein 1-dependent mechanism in mice. Journal of Functional Foods 2015, 19:1-9. doi: http://dx.doi.org/10.1016/j.jff.2015.09.005
  6. Iida T, Moriyama T, Kobata K, Morita A, Murayama N, Hashizume S, Fushiki T, Yazawa S, Watanabe T, Tominaga M. TRPV1 activation and induction of nociceptive response by a non-pungent capsaicin-like compound, capsiate. Neuropharmacology. 2003;44(7):958-67. doi: 10.1016/s0028-3908(03)00100-x.

With regard to the other question, we opted to use just one dose of capsinoids. However, we agree that the use of multiple concentration gradients would enrich the current work, but this suggestion will be applied in future studies in our laboratory in order to investigate the effect of different concentration gradients on obesity and oxidative stress. some studies will be initiated with this proposition. Thanks again.

  1. The fluorescence signal is not visible in the fluorescence picture in Figure 5.

Dear Reviewer, we appreciate the commentary and as requested the Figure 5 was reformulated to improve the fluorescence.

The new Figure 5 is described below:

  1. Why did the rats in the ObCap group have higher MDA levels than the Ob group, while SOD and CAT were elevated in the ObCap group compared to the Ob group? How can this result be explained?

Dear Reviewer, we appreciate the commentary, and we agree that this finding is intriguing. We didn't expect this finding, but there may be an explanation for it. In a study conducted with cancer cells, the authors observed that flavonoids present in grape skins were able to alter the lipid composition of cells, increasing the concentration of polyunsaturated fatty acids (PUFAs) (1). As is scientifically known, under conditions of oxidative stress, free radicals attack lipids containing carbon-carbon double bond(s), especially polyunsaturated fatty acids (PUFAs), increasing the levels of malondialdehyde, one of the indicators of lipid peroxidation (2). Our results showed that the ObCap group had higher MDA levels compared to the Ob. Thus, the hypothesis is that the bioactive compounds we are providing are transforming unsaturated fatty acids into polyunsaturated fatty acids (PUFAs); it should be noted that obesity of current study have more unsaturated fatty acids stored than polyunsaturated ones. In this sense, when we give bioactive compounds there is a change in this proportion, with PUFAs being more dispersed and more susceptible to oxidation. Therefore, if the animals are exposed to an obesogenic diet, becoming obese with high oxidative stress and an increase in PUFAs, more PUFAs will be oxidized, resulting in more MDA. From this situation, there may have been an increase in antioxidant enzymes (SOD and catalase) in an attempt to reverse this increase. Although the lipid profile in the heart was not evaluated in this study, we can suspect that the capsinoids treatment may have provided an increase in PUFAs. In contrast, these high levels provided an increase in the enzymes SOD and catalase, in an attempt to block peroxidation.

  1. Tutino V, Gigante I, Milella RA, De Nunzio V, Flamini R, De Rosso M, Scavo MP, Depalo N, Fanizza E, Caruso MG, Notarnicola M. Flavonoid and Non-Flavonoid Compounds of Autumn Royal and Egnatia Grape Skin Extracts Affect Membrane PUFA's Profile and Cell Morphology in Human Colon Cancer Cell Lines. Molecules 2020;25(15):3352. doi: 10.3390/molecules25153352.
  2. Ayala A, Muñoz MF, Argüelles S. Lipid peroxidation: production, metabolism, and signaling mechanisms of malondialdehyde and 4-hydroxy-2-nonenal. Oxid Med Cell Longev 2014:2014:360438. doi: 10.1155/2014/360438. Epub 2014 May 8.

  1. The conclusion section is too succinct; it is suggested to be added.

Dear Reviewer, we appreciate the commentary and as requested the conclusion section was improved and added in the text in order to add more information. The new conclusion is “The treatment with capsinoids does not prevent the body fat accumulation or promotes improvements in metabolic profile induced by obesity, as well as does not suppress the size of visceral fat. Specifically in heart tissue, capsinoids supplementation does not prevent the process of cardiac remodeling and does not lead to positive modulation of oxidative stress in high-fat diet induced obesity. However, treatment with capsinoids causes a reduction in troponin I and induces antioxidant activity in cardiac tissue (Conclusion, page 14 and Abstract section).

  1. References are not introduced in a uniform format, e.g. line 597, line 232.

Dear Reviewer, we appreciate the commentary and as requested the references were corrected and introduced again with a uniform format.

The corrected references are highlighted and presented throughout the text.

  1. The discussion section repeats the introduction of too much literature, suggest streamlining.

Dear Reviewer, we appreciate the commentary, and as requested the discussion was reduced and there was an attempt to reorganize the writing, as well as avoiding information already described in the introduction.  Thus, we tried to be more consistent and consolidated.

The new discussion was described in the revised manuscript.

  1. Capsaicinoids are potential antioxidants that can reduce obesity-induced oxidative damage and its complications. Why do the results suggest that capsaicin treatment presents the opposite effect in reducing obesity-induced oxidative stress?

Dear Reviewer, we appreciate the commentary, and we believe that the answer to this question was previously reported in question 3, since we explained why we did not observe an antioxidant effect in this study that can reduce obesity-induced oxidative damage and its complications.

  1. The secondary headings in the Materials and Methods section are not sequenced.

Dear Reviewer, we appreciate the commentary and as requested the secondary headings were added and sequenced in the Material and Methods section.

The corrected secondary headings are presented in Material and Methods section and are described below:

  1. Materials and Methods

2.1 Animal care and experimental protocol

2.2 Administration of Capsinoids

2.3 Euthanasia

2.4 Body weight and adiposity

2.5 Visceral adipose tissue morphology

2.6 Comorbidities Associated with Obesity

2.7 Cardiac remodeling and injury

2.8 Analysis of the production of reactive species

2.9 Biomarkers of cardiac oxidants and antioxidants

2.10 Statistical analysis

  1. There are many writing irregularities in the article, such as H2O2 should be revised to H2O2 and O2- to O2- in line 59

Dear Reviewer, we appreciate the commentary, and the writing irregularities were corrected, and the words rewritten in the whole text.

As previously stated, obesity is related to an increase in oxygen reactive species (ROS) in the organism, as superoxide radical (O2•-), hydrogen peroxide (H2O2), hydroxyl radical (•OH), nitric oxide radical (NO•) and peroxynitrite (ONOO) [10]. (Introduction, page 2, third paragraph)

Reviewer 2 Report

Comments and Suggestions for Authors

Capsinoids promotes an increase in enzyme antioxidative 2 activity and prevents the injury cardiac induced by obesity 3 without positively modulate the cardiac oxidative biomarkers

Santos KCC et al

General comments: This study tests the efficacy of capsinoids, a class of plant natural products, shown to benefit those with obesity and complications therein. The authors fed mice either low or high-fat diets, and treated mice in groups of both diets with capsinoids to see what cardiac antioxidant and anti-remodeling impacts these compounds would have. The major issues with this study are 1) the omission of the low-fat diet mice + capsinoids group, and 2) inappropriate use of statistical testing for the groups present. (The authors’ transparency and honest reporting are appreciated.) Also missing are straightforward further measurements of mechanisms and cardiac remodeling, as many are discussed and suggested, and troponin 1 and other cardiac measurements are too minimal to determine cardiac remodeling. Authors are encouraged to add data to this study, as well as apply the correct statistical testing. Minor: small typos throughout.

Introduction:

·       Reactive species should be referred to as reactive oxygen species. Also, why abbreviate as RE instead of ROS?

·       Fatty acids are mentioned in the fifth paragraph, but should be a central part of the obesity background. Please consider moving this up to earlier in the intro.

·       Capsicum should be italicized.

·       Please be more detailed about the capsinoids’ established antioxidant activity. How does it work in the literature?

·       Please expand CVD background to include cardiac endpoints, and move to early in the intro.

Methods:

·       Please detail the sources of lipids in animal diets.

·       It is really important to compare low-fat fed animals + capsinoids with the high-fat fed animals. If you had included this group in the analysis instead of omitting it, this would provide insight into why capsinoids seems to promote ROS and other obesity markers instead of dampening them as expected.

·       Please provide details on which capsinoids were used. They appear to be purchased in aggregate, but which ones are in your test material?

·       And, no solubility issues with water?

·       What do you mean when you say that animals were “heparinized”?

·       Please add detail about how you used Image J. What did the program do to analyze your slides?

·       With three groups, a one-way ANOVA is the best test, not a Student-t test. If you can incorporate the control + capsinoids group, you can do a two-way ANOVA. With just a t-test, we can’t really tell what differences truly exist in your data.

Results:

·       The data have obvious directionality, but without the appropriate statistics, we don’t know how to interpret them.

·       Troponin 1 and LV/whole heart measurements are too limited to characterize remodeling. Why not include RV hypertrophy, collagen and elastin content, or other IHC measures?

·       Figure 5 just shows black images in both the pdf and printed version.

·       I would disagree with the statement mentioning that stimulus for SOD and CAT activity were reduced in both C and Ob groups in Figure 6. A more precise interpretation is that capsinoids are potentiating a climate of elevated ROS, as evidenced by most of Figure 6.

Discussion:

·       This section should be streamlined and cut way back. It is well written, but could definitely be consolidated.

·       Missing is why capsinoids seem to cause elevated ROS. Could this be due to fatty acids components in their chemical structure? Other reasons? I disagree that the activity observed is compensatory.

·       Consider moving paragraphs 8 and 9 to the front.

·       Can you measure TRPV1 receptors, NF-kB, UCP-2 (or 1), and Nrf2? These data would greatly contribute to your story.

Comments on the Quality of English Language

Minor typos.

Author Response

Manuscript number:

Nutrients-3142743

Title:

Capsinoids promotes an increase in enzyme antioxidative activity and prevents the injury cardiac induced by obesity without positively modulate the cardiac oxidative biomarkers

REVIEWER COMMENTS

Reviewer 2

General comments: This study tests the efficacy of capsinoids, a class of plant natural products, shown to benefit those with obesity and complications therein. The authors fed mice either low or high-fat diets, and treated mice in groups of both diets with capsinoids to see what cardiac antioxidant and anti-remodeling impacts these compounds would have. The major issues with this study are 1) the omission of the low-fat diet mice + capsinoids group, and 2) inappropriate use of statistical testing for the groups present. (The authors’ transparency and honest reporting are appreciated.) Also missing are straightforward further measurements of mechanisms and cardiac remodeling, as many are discussed and suggested, and troponin 1 and other cardiac measurements are too minimal to determine cardiac remodeling. Authors are encouraged to add data to this study, as well as apply the correct statistical testing. Minor: small typos throughout.

Introduction:

  1. Reactive species should be referred to as reactive oxygen species. Also, why abbreviate as RE instead of ROS?

Dear Reviewer, we appreciate the commentary, and as requested we rewritten the sentence and added ROS instead of RE. This information is correct.

The correct sentence is “As previously stated, obesity is related to an increase in oxygen reactive species (ROS) in the organism, as superoxide radical (O2•-), hydrogen peroxide (H2O2), hydroxyl radical (•OH), nitric oxide radical (NO•) and peroxynitrite (ONOO−) [10].” (Introduction, page 2, third paragraph)

  1. Fatty acids are mentioned in the fifth paragraph but should be a central part of the obesity background. Please consider moving this up to earlier in the intro.

Dear Reviewer, we appreciate the commentary, but, when the fatty acids were introduced in fifth paragraph, we thought of showing this information in the context of the heart rather than obesity (central part of the obesity background). In this sense, we respectfully believe that the chosen location of this subject could be maintained.

  1. Capsicum should be italicized.

Dear Reviewer, we appreciate the commentary, and as requested Capsicum was italicized.

The new sentence is “In this scenario, studies with capsinoids (CAP) have been used; the CAP consist of alkaloid molecules that are vanillic alcohol esters with fatty acids present in peppers of the Capsicum genus [19-21]. (Introduction, page 2, sixth paragraph, line 7)

  1. Please be more detailed about the capsinoids’ established antioxidant activity. How does it work in the literature?

Dear Reviewer, we appreciate the commentary, and as requested, more details about antioxidant activity of capsinoids were added in the text.

The new sentence is “Regarding antioxidant activity, the literature has shown that compounds belonging to the capsinoids group can exhibit activities such as restoration of gene expression of antioxidant enzymes, protection against lipid oxidation and reduction of 8-hydroxyguanosine (8-OHdG), a marker of oxidative damage in proteins and DNA [28,29]. They may confer a protective and mediating effect against oxidative stress.” (Introduction, page 3, sixth paragraph, line 12)

  1. Ohyama K, Suzuky K. Dihydrocapsiate improved age-associated impairments in mice by increasing energy expenditure. J. Physiol. Endocrinol. Metab. 2017; 313(5): E586-97. https://doi.org/10.1152/ajpendo.00132.2017.
  2. Rosa A, Deiana M, Corona G, Atzeri A, Incani A, Appendino G, Dessi M.A. Protective effect of capsinoid on lipid peroxidation in rat tissues induced by Fe-NTA. Free Radic Res 2005; 39(11): 1155-62. https://doi.org/10.1080/10715760500178094.

  1. Please expand CVD background to include cardiac endpoints, and move to early in the intro.

Dear Reviewer, we appreciate your comment and, as requested, the cardiac endpoints were included earlier in the text. However, we understand that the cardiac changes related to the effects of oxidative stress should remain where they were described originally.

The new sentence is “In the context of cardiovascular diseases (CVDs), these effects are relevant, as they are related to a greater risk of developing left ventricular hypertrophy, hypertension, cardiometabolic disorders, atrial fibrillation, dysfunction and heart failure [9-13].” (Introduction, page 2, second paragraph)

  1. Cavalera M, Wang J, Frangogiannis N.G. Obesity, metabolic dysfunction and cardiac fibrosis: pathophysiologic pathways, molecular mechanisms and therapeutic opportunities. Transl Res 2014; 164(4): 323-35. https://doi.org/10.1016/j.trsl.2014.05.001.
  2. Gadde K.M., Martin C.K., Berthoud H.R., Heymsfield S.B. Obesity: pathophysiology and management. J Am Coll Cardiol 2018, 71(1):69-84. https://doi.org/10.1016/j.jacc.2017.11.011.
  3. Ilkun O, Boudina S. Cardiac dysfunction and oxidative stress in the metabolic syndrome: an update on antioxidant therapies. Curr Pharm Des 2013; 19(27):4806-17. http://dx.doi.org/10.2174/1381612811319270003.
  4. Koenen M, Hill M.A, Cohen P, Sowers J.R. Obesity, adipose tissue and vascular dysfunction. Circ Res 2021;128(7):951-68. https://doi.org/10.1161/CIRCRESAHA.121.318093.
  5. Powell-Wiley T.M., Poirier P, Burke L.E., et al. American Heart Association Council on Lifestyle and Cardiometabolic Health; Council on Cardiovascular and Stroke Nursing; Council on Clinical Cardiology; Council on Epidemiology and Prevention; Stroke Council. Obesity and Cardiovascular Disease: A Scientific Statement From the American Heart Association. Circulation 2021;143(21): e984-e1010. https://doi.org/10.1161/CIR.0000000000000973.

Methods:

  1. Please detail the sources of lipids in animal diets.

Dear Reviewer, we appreciate the commentary, and as requested the sources of lipids were detailed in the text.

The standard diet was based on the AIN 93 recommendations, being composed of corn starch, casein, dextrinized starch, sucrose, soy oil, microcrystalline cellulose, mineral mix, vitamin mix, L-cystine, BHT and bitartrate choline (Pragsoluções biociências ®, São Paulo, Brazil). This diet was composed of 9.45 % lipids (100 % soy oil), 75.66 % carbohydrates and 14.89 % proteins (Pragsoluções biociências ®, São Paulo, Brazil). For the composition of the high-fat diet, the same recommendations were followed, but with the addition of lard (Pragsoluções biociências ®, São Paulo, Brazil). The high-fat diet was composed of 45.33 % lipids (11.5 % soy oil and 88.5 % pork lard), 40.29 % carbohydrates and 14.38 % proteins (Pragsoluções biociências ®, São Paulo, Brazil). (Material and Methods, Animal care and experimental protocol, page 3, first paragraph)

  1. It is really important to compare low-fat fed animals + capsinoids with the high-fat fed animals. If you had included this group in the analysis instead of omitting it, this would provide insight into why capsinoids seems to promote ROS and other obesity markers instead of dampening them as expected.

Dear Reviewer, we appreciate the commentary, and agree that the results from CCAP would enrich the current work, but our main objective was to investigate the effects of treatment with capsinoids on obesity. In this sense, the CCap group would not benefit us in the discussion of obesity, since we would have to address its effect on standardized diets and in the absence of obesity. However, this suggestion will be applied in future studies in our laboratory in order to investigate and to compare the effect of low-fat fed animals + capsinoids with the high-fat fed animals on obesity and oxidative stress. Some studies will be initiated with this proposition. Thanks again.

  1. Please provide details on which capsinoids were used. They appear to be purchased in aggregate, but which ones are in your test material?

Dear Reviewer, we appreciate the commentary, and as requested the details of capsinoids were determined and added in the text. The compound used for the treatment was formulated by a company and was purchased ready for use, consisting of a mixture of the capsinoids capsiate, dihydrocapsiate and nordihydrocapsiate. It has already been corrected in the text as sentence described below:

ObCap group was supplemented daily with capsinoids (CAP; Infinity Pharma, Brazil) (capsiate, dihydrocapsiate and nordihydrocapsiate) by orogastric gavage (10 mg of capsinoids/kg of BW diluted in 1 ml of water/kg of body weight) for 8 weeks. (Material and Methods, page 4, first paragraph)

  1. And, no solubility issues with water?

Dear Reviewer, we appreciate the commentary, but we didn’t have problems with the solubility. Although some studies show that capsinoids have poor water solubility or are not soluble in water (1-3), in the current study we did not encounter this problem, since the capsinoids dilution was excellent and was soluble in water as this study (4); and we applied it by orogastric gavage. Cantrell and Jarret (5) evaluated the efficacy of various organic solvents for the extraction of capsinoids from dried Capsicum annuum fruit. Among the organic solvents evaluated, pentane appeared to provide a good combination of both recovery and purity. The results were that 26.3% (wt/wt) capsiate and 19.4% (wt/wt) dihydrocapsiate for a combined capsinoids yield of 45.7% (wt/wt). A third step, involving a rapid hp20ss chromatography column using a water/acetonitrile gradient, resulted in a combined capsinoids yield of 96.6% (wt/wt).

  1. Basith S., Cui M., Hong S., Choi S. Harnessing the Therapeutic Potential of Capsaicin and Its Analogues in Pain and Other Diseases. Molecules 2016 23;21(8):966. doi: 10.3390/molecules21080966.
  2. Cho S., Moon H.I., Hong G.E., Lee C.H., Kim J.M., Kim S.K. Biodegradation of capsaicin by Bacillus licheniformis SK1230. J Korean Society for Applied Biological Chem 2014, 57: 335-9.
  3. Rollyson W.D., Stover C.A., Brown K.C., Perry H.E., Stevenson C.D., McNees C.A., Ball J.G., Valentovic M.A., Dasgupta P. Bioavailability of capsaicin and its implications for drug delivery. J Control Release 2014:196:96-105. doi: 10.1016/j.jconrel.2014.09.027. Epub 2014 Oct 12.
  4. Oğuzkan S.B. Extraction of Capsinoid and its Analogs from Pepper Waste of Different Genotypes. Natural Product Communications 2019: 1–5. doi: 10.1177/1934578X19865673.
  5. Cantrell C.L., Jarret R.L. Bulk Process for Enrichment of Capsinoids from Capsicum Fruit. Processes 2022, 10(2), 305; https://doi.org/10.3390/pr10020305

  1. What do you mean when you say that animals were “heparinized”?

Dear Reviewer, we appreciate the commentary, and as requested the explanation for use of heparin in our rats is related to cardiomyocyte technique (1). Heparin in animals used in the cardiomyocyte technique enables a better fixation to optimize perfusion-fixation. Heparin as an anticoagulant can be used in vivo or in vitro (1). In some protocols, heparin has been added to the cannulation solution for use (2). However, instead of using heparin in the cannulation solution, we injected heparin before harvesting the heart, which can let the anticoagulant blood flow into the coronary to prevent the formation of embolism. In addition, blood collection tubes containing heparin, which stabilizes the red blood cell membranes, are used for several specialized chemistry tests as the biochemistry (lipid profile).

  1. Xiangang Tian, Meng Gao, Anqi Li, Bilin Liu, Wenting Jiang, Guohua Gong. Protocol for Isolation of Viable Adult Rat Cardiomyocytes with High Yield. STAR Protocols 2020, 1(2): 100045. https://doi.org/10.1016/j.xpro.2020.100045
  2. Plačkić J, Kockskämper J. Isolation of Atrial and Ventricular Cardiomyocytes for In Vitro Studies. Methods Mol Biol. 2018;1816:39-54. doi: 10.1007/978-1-4939-8597-5_3.

  1. Please add detail about how you used Image J. What did the program do to analyze your slides?

Dear Reviewer, we appreciate your commentary. As requested, we detailed who we used the Image J. To the analysis of the morphology of visceral adipose tissue, this software was used to quantify the area of ​​adipocytes. Thus, after the images of the slides had been captured by the optical microscope, the program was used to identify and convert the area of ​​adipocytes considering the scale in µm in each field of the slides. It has already been corrected in the text as sentence described below:

After capturing the images, Image J Pro-Plus® (Media Cybernetics, Silver Spring, Maryland, USA) was used to identify and convert the area of ​​adipocytes considering the scale in µm in each field of the slides. (Material and Methods, Visceral adipose tissue morphology, page 4, line 10)

For the analysis of the production of reactive species, Image J was also used. In this case, after the photos had been captured using the confocal fluorescence microscope, the program was used to quantify the intensity of the fluorescence induced by DHE, and consequently, to quantify the production of reactive species in the left ventricle. The reading of this intensity was performed in arbitrary units. It has also been corrected in the text as sentence described below:

After capturing the fluorescence images, specific software (Image J Pro-Plus®, Media Cybernetics, Silver Spring, Maryland, USA) was used to quantify the fluorescence intensity on the slides and, consequently, to quantify the production of reactive species in the left ventricle. Quantification was performed in arbitrary units and the analysis of 4 fields per sample were performed. (Material and Methods, Visceral adipose tissue morphology, page 4, line 10)

  1. With three groups, a one-way ANOVA is the best test, not a Student-t test. If you can incorporate the control + capsinoids group, you can do a two-way ANOVA. With just a t-test, we can’t really tell what differences truly exist in your data.

Dear Reviewer, we appreciate the commentary, and we totally agree that a one-way ANOVA would be the best test, however, I don't have the CCap group to evaluate the factors in isolation (capsinoids or obesity) or their interaction. Furthermore, as we only wanted to assess the effects of capsinoids on obesity and oxidative stress, and we only compared the Ob versus ObCap groups, the most appropriate test is the Student's t-test for independent samples. Other explanation for not using ANOVA refers to the possible comparisons, for example, not comparing ObCap and C, because what factor would be involved in changing the variables. In this context, we have used and presented group C graphically only to show the effect of obesity, however, if there is a need and the reviewers think it would be better to remove this comparison, we can redo the graphs with just the 2 groups (Ob vs. ObCap).

Results:

  1. The data have obvious directionality, but without the appropriate statistics, we don’t know how to interpret them.

Dear Reviewer, we appreciate the commentary, and we agree that an incorporation of the control + capsinoids group would enrich our work, allowing us to use a two-way ANOVA and a more in-depth discussion on the subject. However, as explained above, our main objective was to investigate the effects of capsinoids on cardiac oxidative stress in obese rats induced by a high-fat diet. I would also stress that we used the appropriate statistics, and our findings are consistent. We thank you for your suggestion, which will be applied in future studies by our laboratory.

  1. Troponin 1 and LV/whole heart measurements are too limited to characterize remodeling. Why not include RV hypertrophy, collagen and elastin content, or other IHC measures?

Dear Reviewer, we appreciate the commentary, and we agree that RV hypertrophy, collagen and elastin content, or other IHC measures, would enrich our work, allowing a better characterization of cardiac remodeling. However, as we used the animals' hearts in the cardiomyocyte technique, we were unable to obtain other samples for histological analysis. These measurements were planned in the study's initial proposal, but many animals would have been needed and our Brazilian Ethics Committee does not allow the use of large numbers of rats. However, several studies have verified the macroscopic pathological cardiac remodeling by evaluation of total mass of the heart, left ventricle (LV) and their relationship with tibia length (1,2), as well as troponin I as a marker of cardiac injury (3,4). 

  1. Koch, V.; Weber, C.; Riffel, J.H.; Buchner, K.; Buss, S.J.; Hein, S.; Mereles, D.; Hagenmueller, M.; Erbel, C.; März, W.; Booz, C.; Albrecht, M.H.; Vogl, T.J.; Frey, N.; Hardt, S.E.; Ochs, M. Impact of Homoarginine on Myocardial Function and Remodeling in a Rat Model of Chronic Renal Failure. J. Cardiovasc. Pharm. T. 2022, v.27. p.1-12. https://doi.org/10.1177/10742484211054620.
  2. Bevere, M.; Morabito, C.; Guarnieri, S.; Mariggiò, M.A. Mice lacking growth-associated protein 43 develop cardiac remodeling and hypertrophy. Histochem. Cell. Biol. 2022, v.157, n.5, p.547-556. https://doi.org/10.1007%2Fs00418-022-02089-x.
  3. York, M.; Scudamore, C.; Brady, S.; Chen, C.; Wilson, S.; Curtis, M.; Evans, G.; Griffiths, W.; Whyman, M.; Williams, T.; Turton, J. Characterization of Troponin Responses in Isoproterenol-Induced Cardiac Injury in the Hanover Wistar Rat. Toxicol. Pathol. 2007, v.35, n.4, p.606-617. https://doi.org/10.1080/01926230701389316.
  4. Atas, E.; Kismet, E.; Kesik, V.; Karaoglu, B.; Aydemir, G.; Korkmazer, N.; Demirkaya, E.; Karslioglu, Y.; Yurttutan, N.; Unay, B.; Koseoglu, V.; Gokcay, E. Cardiac troponin-I, brain natriuretic peptide and endothelin-1 levels in a rat model of doxorubicin-induced cardiac injury. J. Cancer Res. Ther. 2015. v.11 n.4, p.882-886. http://dx.doi.org/10.4103/0973-1482.144636.

  1. Figure 5 just shows black images in both the pdf and printed version.

Dear Reviewer, we appreciate the commentary and as requested the Figure 5 was reformulated to improve the fluorescence.

The new Figure 5 is described below:

  1. I would disagree with the statement mentioning that stimulus for SOD and CAT activity were reduced in both C and Ob groups in Figure 6. A more precise interpretation is that capsinoids are potentiating a climate of elevated ROS, as evidenced by most of Figure 6.

Dear Reviewer, we appreciate the commentary, and we totally agree that capsinoids are potentiating a climate of elevated ROS. Therefore, we removed the last sentence and added this information in the text.

The new sentence is “Finally, it was demonstrated that the SOD and CAT were elevated in ObCap than Ob group (+62.5% and +636.4%, respectively), but no differences were observed between groups C and Ob (Figures 6E and 6F). Thus, these results showed that capsinoids are potentiating a climate of elevated ROS.” (Results, pages 10 and 11, last paragraph).

Discussion:

  1. This section should be streamlined and cut way back. It is well written, but could definitely be consolidated.

Dear Reviewer, we appreciate the commentary, and as requested the discussion was reduced and there was an attempt to reorganize the writing, as well as avoiding information already described in the introduction.  Thus, we tried to be more consistent and consolidated.

  1. Missing is why capsinoids seem to cause elevated ROS. Could this be due to fatty acids components in their chemical structure? Other reasons? I disagree that the activity observed is compensatory.

Dear Reviewer, we appreciate the commentary, and we agree that this finding is intriguing. We didn't expect this finding, but a possible explanation for the elevated ROS induced by capsinoids in obesity is related to polyunsaturated fatty acids (PUFAs). In a study conducted with cancer cells, the authors observed that flavonoids present in grape skins were able to alter the lipid composition of cells, increasing the concentration of polyunsaturated fatty acids (PUFAs) (1). As is scientifically known, under conditions of oxidative stress, free radicals attack lipids containing carbon-carbon double bond(s), especially polyunsaturated fatty acids (PUFAs), increasing the levels of malondialdehyde, one of the indicators of lipid peroxidation (2). Our results showed that the ObCap group had higher MDA levels compared to the Ob. Thus, the hypothesis is that the bioactive compounds we are providing are transforming unsaturated fatty acids into polyunsaturated fatty acids (PUFAs); it should be noted that obesity of current study have more unsaturated fatty acids stored than polyunsaturated ones. In this sense, when we give bioactive compounds there is a change in this proportion, with PUFAs being more dispersed and more susceptible to oxidation. Therefore, if the animals are exposed to an obesogenic diet, becoming obese with high oxidative stress and an increase in PUFAs, more PUFAs will be oxidized, resulting in more MDA. From this situation, there may have been an increase in antioxidant enzymes (SOD and catalase) in an attempt to reverse this increase. Although the lipid profile in the heart was not evaluated in this study, we can suspect that the capsinoids treatment may have provided an increase in PUFAs. In contrast, these high levels provided an increase in the enzymes SOD and catalase, in an attempt to block peroxidation.

  1. Tutino V, Gigante I, Milella RA, De Nunzio V, Flamini R, De Rosso M, Scavo MP, Depalo N, Fanizza E, Caruso MG, Notarnicola M. Flavonoid and Non-Flavonoid Compounds of Autumn Royal and Egnatia Grape Skin Extracts Affect Membrane PUFA's Profile and Cell Morphology in Human Colon Cancer Cell Lines. Molecules 2020;25(15):3352. doi: 10.3390/molecules25153352.
  2. Ayala A, Muñoz MF, Argüelles S. Lipid peroxidation: production, metabolism, and signaling mechanisms of malondialdehyde and 4-hydroxy-2-nonenal. Oxid Med Cell Longev 2014:2014:360438. doi: 10.1155/2014/360438. Epub 2014 May 8.

  1. Consider moving paragraphs 8 and 9 to the front.
  2. Dear Reviewer, we appreciate the commentary, and as requested the discussion was reduced and there was an attempt to reorganize the writing, as well as avoiding information already described in the introduction. Thus, we tried to be more consistent and consolidated.

  1. Can you measure TRPV1 receptors, NF-kB, UCP-2 (or 1), and Nrf2? These data would greatly contribute to your story.

Dear Reviewer, we appreciate the commentary, and we totally agree that the measurement of TRPV1 receptors, NF-kB, UCP-2 (or 1), and Nrf2 would enrich our work, allowing the understanding of mechanism related to capsinoids. However, we no longer have samples available for these analyses. It's worth noting that other studies in our laboratory plan to analyze these markers. Thank you very much for your suggestion, which will be applied in future studies in our laboratory.

Round 2

Reviewer 1 Report

Comments and Suggestions for Authors

NO COMMENT IN THIS VERSION, AUTHORS REVISED WELL ACCORDING TO MY SUGGESTIONS

Author Response

REVIEWER COMMENTS

Reviewer 1

Journal: Nutrient

 Comments and Suggestions for Authors

NO COMMENT IN THIS VERSION, AUTHORS REVISED WELL ACCORDING TO MY SUGGESTIONS

Dear Reviewer, we appreciate the commentary, and we thank you for your comments, as they have enriched our work.

Reviewer 2 Report

Comments and Suggestions for Authors

The authors have made substantial edits to the current manuscript and have taken the reviewer comments seriously, which is appreciated. The authors suggest removing a group from the analysis and only presenting data on two groups, with a student t-test for analysis. This modification is the only way to correct the control group and statistical issues, and is highly encouraged.

Comments on the Quality of English Language

Only minor typos and misspellings.

Author Response

Manuscript number:

Nutrients-3142743

Title:

Capsinoids promotes an increase in enzyme antioxidative activity and prevents the injury cardiac induced by obesity without positively modulate the cardiac oxidative biomarkers

REVIEWER COMMENTS

Reviewer 2

Comments and Suggestions for Authors

The authors have made substantial edits to the current manuscript and have taken the reviewer comments seriously, which is appreciated. The authors suggest removing a group from the analysis and only presenting data on two groups, with a student t-test for analysis. This modification is the only way to correct the control group and statistical issues, and is highly encouraged.

Dear Reviewer, we appreciate the commentary and, as requested only the results of the obese groups (Ob and ObCap) were analyzed and presented. These sections were reformulated, and they are presented in the revised manuscript.